# Disc Degeneration and Cervical Spine Intervertebral Motion: A Cross-Sectional Study in Patients with Neck Pain and Matched Healthy Controls

**DOI:** 10.3390/jfmk9010055

**Published:** 2024-03-19

**Authors:** Jonathan Branney, Alexander Breen, Alister du Rose, Philip Mowlem, Alan Breen

**Affiliations:** 1Faculty of Health and Social Sciences, Bournemouth University, Bournemouth BH12 5BB, UK; jbranney@bournemouth.ac.uk; 2Faculty of Science and Technology, Bournemouth University, Bournemouth BH12 5BB, UK; alexbreen@bournemouth.ac.uk (A.B.); abreen4@bournemouth.ac.uk (A.B.); 3School of Chiropractic, AECC University College, Bournemouth BH5 2DF, UK; 4Department of Radiology, University Hospitals Dorset, Poole BH15 2JB, UK; pmowlem@yahoo.co.uk

**Keywords:** disc degeneration, neck pain, fluoroscopy, cervical spine

## Abstract

While neck pain can be defined in clinical terms, in most cases the underlying pathophysiology is largely unknown. Regional cervical spine range of motion is often found to be reduced in patients with neck pain compared to persons without pain although it is not clear if the decreased range is cause or effect. Less is known about the role of intervertebral kinematics and how that might be related to the presence of disc degeneration. In this study, the prevalence of intervertebral disc degeneration and continuous cervical intervertebral motion were both measured utilizing quantitative fluoroscopy (QF) in patients with subacute or chronic neck pain (*n* = 29) and gender-matched healthy controls (*n* = 30). A composite disc degeneration (CDD) score was calculated for each participant from the first, neutral, lateral fluoroscopic image. Intervertebral motion sharing parameters of motion-sharing inequality (MSI) and motion-sharing variability (MSV) were derived from the active cervical motion sequences obtained while patients were seated. The objective was to determine if average age, CDD, MSI, and MSV values were correlated and if there were differences in these variables between the neck pain group and the healthy control group. Correlation analysis was conducted for age, CDD, MSI, and MSV in each group. Age was moderately correlated with MSV in cervical spine extension in patients only (r = 0.63, *p* < 0.001). There were no significant differences in the prevalence of disc degeneration (CDD) between patients, who had on average mild pain and related disability, and healthy controls (median CDD 2 both groups, *p* = 0.94). There were also no significant differences in either flexion or extension intervertebral motion-sharing inequality or variability (MSI or MSV) between groups as measured during active cervical motion.

## 1. Introduction

Neck pain ranks as the fourth leading cause of years lived with disability [1]. Despite neck pain having a significant global burden, accurate diagnosis and, therefore, accurate targeting of treatment remains elusive in most cases [2]. Surgical intervention can be indicated in pathological causes of neck pain such as infection or cancer with an associated risk of spinal cord or airway compression [3] or after cervical spine trauma [4]. In the absence of serious pathology though, the role of surgery in managing neck pain is uncertain with evidence suggesting a lack of superiority compared to conservative management [5].

Rather than pathological, most cases of neck pain are considered at least partially ‘mechanical’ in nature [3]. This is based on the clinical finding that pain is made worse by neck movement and/or by provocative orthopaedic testing [4]. It is thus inferred that the source of pain is one or more of the innervated, and therefore potentially pain-producing, structures of the cervical spine [5]. The intervertebral disc has attracted attention as a potentially key pain-producing structure; however, disc degeneration as a cause of neck pain remains controversial due to degeneration being common in people without neck pain as assessed by static imaging [6]. While pain provocation with cervical discography might be of value in carefully selected patients, the evidence base is weak [7].

While the utility of static radiographic imaging might be limited for mechanical neck pain, intervertebral motion imaging with quantitative fluoroscopy (QF) has the potential to offer additional information that may be of diagnostic value. Most work in this area has been in the lumbar spine. In a study by Mellor et al. (2014), QF was used to measure intervertebral motion sharing, and this motion was found to be more variable in patients with chronic non-specific low back pain than in controls [8]. However, that study did not measure disc degeneration.

There are a few studies that have investigated continuous intervertebral motion in the cervical spine [9,10,11] but only one appears to have assessed the potential contribution of disc degeneration [12]. In that pilot study, 5 blinded surgeons reliably identified from cinematographic recordings that the cervical intervertebral motion of 20 healthy controls was different from that of 10 pre-operative patients with cervical intervertebral disc degenerative disease [12].

A more recent study in chronic non-specific low back pain patients explored the potential relationship between disc degeneration and intervertebral motion sharing as measured by QF. In that study, Breen and Breen (2018) analysed proportional intervertebral motion by way of both passive recumbent and active weight-bearing motion sharing inequality (MSI), which represented the unevenness of restraint between segments, and motion sharing variability (MSV), which represented the unevenness of control [13]. The degree of disc degeneration was found to be correlated with uneven proportional motion sharing between segments [13]. The relationship between disc mechanics and uneven intervertebral motion (MSI and MSV) suggests a plausible mechanism for generating pain, but this has not been investigated in the cervical spine.

The objectives of this study were to determine the relationship between intervertebral motion-sharing inequality/variability, age, and disc degeneration in patients with neck pain and healthy controls during active, weight-bearing flexion and extension, and to identify differences, if present, between the two groups.

## 2. Materials and Methods

This was a cross-sectional observational study of disc degeneration and intervertebral motion sharing in the cervical spine in patients with neck pain and matched healthy controls. This involved the secondary analysis of fluoroscopic image sequences obtained in an earlier study. The report is described in accordance with the STROBE checklist for cross-sectional studies [14].

### 2.1. Participants

Twenty-nine patients with subacute or chronic neck pain (twenty-one female) and thirty age and gender-matched healthy controls were recruited from a musculoskeletal outpatient clinic between August 2011 and April 2013 for a previous study [9].

Participants could be included if they were male or female, aged 18–70, had no medical radiation exposure greater than 10 mSv in the previous 2 years, and were not pregnant. Healthy controls needed to have no current neck pain, dizziness or vertigo and no activity-limiting neck pain lasting more than 24 h in the past year. Patients needed to have at least 2 weeks’ mechanical neck pain rated 3 or more on the 11-point Numerical Rating Scale [15] as a measure of pain severity, to be included. Patients were not excluded on the basis of any pain medication they were taking.

Patients also completed the Neck Disability Index [16] as a measure of pain-related disability and EuroQuol-5D-5L [17] as a quality-of-life measure. Exclusion criteria for patients included depression, litigation/compensation pending, or evidence of central hypersensitivity as assessed by pressure algometry. All participants gave informed consent. The study was conducted in accordance with the Declaration of Helsinki and the study protocol was approved by the UK National Research Ethics Service (South West—Cornwall and Plymouth, REC reference 11/SW/0072).

### 2.2. Image Acquisition

The quantitative fluoroscopy (QF) acquisition procedure for the cervical spine is explained in detail elsewhere [9]. In brief, participants were seated with bracing rods comfortably positioned against the mid-thoracic spine and the sternum to promote a fixed trunk position (Figure 1).

A face rest rigidly attached to a motorised vertical motion frame (Atlas Clinical Ltd., Lichfield, UK) was brought to touch the participant’s cheeks with instructions to follow the movement of the face rest during fluoroscopy. The face rest did not impose any motion or support the weight of the participant’s head at any time. Participants performed flexion as the face rest rotated downwards, and extension as it moved upwards, keeping their faces’ gently in touch with the face rest throughout the motion (Figure 2).

The motion frame controller was set to reproduce participants’ maximum attainable flexion and extension ranges as established using the CROM instrument (Performance Attainment Associates Inc., St Paul, MN, USA). The computer-controlled motion-frame rate was set at 3° per second as per international consensus [18]. Prior to measurement, participants were instructed to warm up with five neck flexion–extension repetitions. Flexion and extension were imaged as two separate motion sequences from neutral. Imaging sequences were obtained from a Siemens Arcadis Avantic VC10A digital fluoroscope (Siemens GMBH, Nuremberg, Germany) at a rate of 15 frames per second.

### 2.3. Image Analysis

Following motion imaging, the DICOM files were exported to a computer workstation for analysis. Manual registration was used on the first image to identify the individual vertebral bodies (Figure 3). Then, in all subsequent frames of the motion sequence, these vertebrae were tracked with bespoke tracking codes written in Matlab (V2013—The Mathworks Inc., Natick, MA, USA).

One operator (JB) analysed the anonymised image sequences from which intervertebral angular rotation data were obtained. The accuracy and repeatability of this method were determined in an earlier calibration study using a model consisting of a moveable human cervical segment (C4–5) with a digital inclinometer attached. Motion measured by the digital inclinometer as the calibration model was moved by a motor was compared to that measured simultaneously by QF. Accuracy was reported as 0.21° in flexion and 0.34° in extension [9]. For repeatability, intraclass correlation coefficients (ICC) and the standard error of measurement (SEM) were calculated [19]. Inter-observer repeatability in flexion ranged from ICC 0.97–0.99 and SEM 0.3–0.6°, and in extension ICC 0.92–0.97 and SEM 0.8–1.1° [9].

From the C2/3 to C5/6 angular rotational data, a second operator (AxB) calculated the average difference between the minimum and maximum contributions across all motion segments for each frame throughout the motion sequence. This was used to derive two motion parameters for each participant in each direction to express the evenness of the intervertebral motion: motion-sharing inequality (MSI), the mean of the range of proportional intervertebral motion, and motion-sharing variability (MSV), the variance of this range. Figure 4 provides an illustration of how the MSI and MSV were calculated for all participants, with the example of one representative QF image sequence obtained from one participant during cervical flexion and return (Figure 4A–D).

In Figure 4A are the absolute intervertebral rotations in degrees (y-axis) shown against the corresponding image from the sequence as a percentage (x-axis) to best indicate flexion and return. The black line represents the total range contributed to by all segments. Figure 4B shows the proportion contributed to the total range by each segment. The maximum and minimum proportional intervertebral motion at each image is then shown in Figure 4C. The range between each maximum and maximum is calculated and from that, the mean of the range of proportional intervertebral motion, as shown in Figure 4D. The mean of that range is MSI (motion sharing inequality), the standard deviation of that range is MSV (motion sharing variability).

MSI represents the unevenness in restraint between segments and was calculated as follows:$$MSI=\frac{\sum_{i=1}^{N}{fRC}_{i}}{N}$$ where ${fRC}_{i}$ is the average of the range of proportional intervertebral motion across the (*N*) images of the motion sequence (Figure 4D). The mean (standard deviation) number of images per sequence (flexion and extension) across all participants was 290 (38) images. *MSV* represents the unevenness of control and was calculated as follows:$$MSV=\sqrt{\frac{\sum_{i=1}^{N}({fRC}_{i}{-MSI)}^{2}}{N}}$$

A full account of the derivation and calculation of these parameters has been published previously [13]. The inter-observer repeatability of *MSI* has been reported as ICC 0.93 (0.86–0.97), and for *MSV*, ICC 0.55 (0.24–0.75), SEM 0.024 for both parameters, based on 2 investigators analysing lumbar spine flexion motion from thirty healthy controls [20].

The initial image of each flexion imaging sequence with the cervical spine in neutral was assessed for disc degeneration using the Kellgren and Lawrence rating scale [21] by a chiropractor (AD) and, separately, by a radiographer (PM), both of whom were trained to interpret radiographs. This scale is a common method of classifying the severity of osteoarthritis from imaging providing a score from 0 (no radiological findings of osteoarthritis) through to 4 (large osteophytes, marked joint space narrowing, severe sclerosis and definite deformity of bone contour). Where required differences were resolved through agreement. The scores of 0–4 for each level were added together to give a composite disc degeneration score (CDD) for each participant [13]. While other scales are commonly used to grade disc degeneration, the Kellgren and Lawrence system remains an acceptable grading system for lateral cervical views [22]. The use of this scale was to facilitate comparison to a similar lumbar study that used this grading system [13].

### 2.4. Statistical Analysis

Weighted Kappa and intra-class correlation coefficients (ICC) were calculated to determine inter-rater agreement and reliability for radiographically identifying disc degeneration. Normality of the distributions of continuous data was assessed with the Shapiro–Wilk test. Discrete data were analysed for differences with the Fisher’s exact test. Differences in means of normally distributed data between groups were analysed using the unpaired Student’s *t*-test. Differences in medians of non-normal distributions between groups were analysed with the Mann–Whitney U test. Correlations were made with the Spearman’s correlation coefficient and interpreted as follows: 0.00–0.10, negligible; 0.10–0.39, weak; 0.40–0.69, moderate; 0.70–0.89, strong; 0.90–1.00, very strong [23].

## 3. Results

The weighted Kappa for CDD scores was 0.78 (95% CI 0.68–0.88, *p* < 0.001) and the ICC was 0.79 (95% CI 0.66–0.87, *p* < 0.001) suggesting substantial inter-rater agreement and reliability [24]. As shown in Table 1, regional cervical spine range of motion was significantly reduced in patients in both flexion and extension. There were no other significant between-group differences in participant characteristics or CDD scores.

Patients on average reported mild pain and disability and high scores for quality of life (Table 1). The CDD score given is out of a maximum of 16 (maximum grading of 4 for each of 4 levels from C2/3 to C5/6) as the motion of C7 and T1 vertebrae could not be consistently tracked in participants due to radiographic superimposition. The results are shown in Table 2.

The severity of degeneration at C6/7 and C7/T1, as for all other levels, was very similar between groups. In healthy controls, 5 out of 12 C6/7 degenerated discs were classed as Kellgren–Lawrence Grade 1, 5 at Grade 2, 1 at Grade 3, and 1 at Grade 4. In patients, 7 out of 11 degenerated discs were classed as Kellgren–Lawrence Grade 1, 2 at Grade 2, and 2 at Grade 3. Degenerated C7/T1 discs were all classed as Kellgren–Lawrence Grade 1 in both groups. While the patient group had a higher proportion of degenerated discs (62 out of 145 (43%)) versus healthy controls (55 out of 150 (37%)), this difference was not significant (*p* = 0.341). Also presented in Table 2 are the intervertebral ranges of motion by level and direction. There were no significant differences between patients with neck pain and healthy controls.

Table 3 presents the median MSI and MSV for each group and shows there were no significant differences in either flexion or extension between groups.

Age was moderately correlated with CDD score for healthy controls only, as shown in Table 4. There was no correlation between CDD and MSI or MSV in patients or controls. However, this table also shows a moderate correlation between age and MSV in extension, in patients only. MSI was moderately correlated with MSV in both patients and controls in flexion, but only in patients for extension.

## 4. Discussion

This study did not detect a difference in the prevalence of cervical spine intervertebral disc degeneration or intervertebral motion-sharing inequality or variability (MSI or MSV) between patients with neck pain and healthy controls, as measured by quantitative fluoroscopy (QF). A similar QF study in the lumbar spine also arrived at the same findings between patients with chronic low back pain and controls when participants were imaged during active weight-bearing motion [13]. That study did, however, report significantly higher MSI values in patients (*n* = 10) versus controls (*n* = 10) during passive flexion motion. That suggested the presence of uneven restraint from the passive elements of the spine during motion, which could put excessive demands on the spinal musculature, which may in turn contribute to muscle pain over time [25].

Alternatively, or in addition, excess intervertebral laxity might cause greater acceleration of the segmental levels involved relative to other levels, possibly then generating pain from ligaments and/or discs [26,27]. While this present study utilised a motorised face-rest to control the speed at which participants flexed or extended their necks, they were still ultimately in control of their own motion thus it was not possible to disaggregate active, passive, or neural spinal subsystems [28]. There might be important passive motion differences in patients with neck pain, but a system to measure continuous cervical intervertebral motion passively is not currently available.

It is possible that intervertebral motion differences are more likely to be found in patients with more severe pain and disability such as those with traumatic neck pain or who have had cervical spine surgery. In a study utilising bi-planar radiography with CT reconstruction, patients with single-level anterior arthrodesis (*n* = 6) had significantly increased motion contributions across the total C2–C7 flexion–extension range from the two levels adjacent to the C5/6 arthrodesis compared to controls (*n* = 18) [29]. Such motion differences might contribute to accelerated adjacent degenerative changes [30]. These findings contrast with those from a study of 46 cervical fusion patients where no differences were found for rotation or translation at segments adjacent to fused levels from end-range flexion–extension static radiographs [31]. If motion differences are to be detected, it is more likely to be with methods that measure continuous intervertebral motion, and in patients with more severe degenerative changes than those in the present study [12].

The risk of cervical disc degeneration increases with age [30], and in the present study, the cumulative disc degeneration (CDD) score correlated with age; however, this reached statistical significance in controls only. Similar to what was reported from the lumbar spine study by Breen and Breen (2018), a positive correlation was noted between age and MSV in patients only. In this study, the relationship was in extension only, while the lumbar spine study identified the correlation in flexion (extension was not measured) [13]. This suggests that the older the patients with neck pain, the more unevenness of control was present. For this reason, it might be that motor control exercises should be particularly targeted at older patients, but this is speculative and warrants further investigation.

The unevenness of control may partially be explained by pain intensity, which moderately correlated with CDD score (Rho = 0.45, *p* = 0.015). (CDD scores were not correlated with disability (NDI) or quality of life (EQ-5D-5L) in this cohort of largely mild disability neck pain patients). This accords with findings summarised in a recent systematic review, which state that patients with moderate to severe neck pain have altered motor control compared to those without neck pain [32]. Despite this, MSV was not greater in patients versus controls in this present study, which may have been due to the stabilising effects of neck muscle activation during the controlled but active neck motion. Future studies could explore the relationship between neck muscle activity and cervical intervertebral motion with simultaneous recording of QF and EMG [33].

This study did utilise a small convenience sample; however, findings were largely in accordance with previous work suggesting similar conclusions would be reached with a larger sample. Degenerated C2/3 discs were recorded with a higher frequency than other levels which was unexpected when compared to the findings of a large MRI cross-sectional study [34]. However, all but two of the C2/3 degenerated discs were graded 1 on the Kellgren–Lawrence scale. In contrast, there was a greater severity of degeneration in the lower cervical spine in both groups, especially at C5/6 and C6/7 (Table 2), as might be expected [34].

Regional cervical range of motion was reduced in patients with neck pain compared to healthy controls which is a common finding [35]. However, intervertebral angular ranges were not significantly different between the groups. It could be that differences in motion occurred at levels which could not be measured (C0/1, C6/7, C7/T1). Alternatively, patients perhaps felt more confident in moving despite the pain while guided by a face-rest and motion frame, which was calibrated to their individual attainable regional motion range, compared to free-bending. It would have been advantageous to measure cervical intervertebral motion passively in addition to actively, to disaggregate the influence of the spinal musculature, but a passive system was not available.

The lead author was not blinded as to whether an individual’s data was from a patient with neck pain or a healthy control which created a risk of bias when interpreting the data analysis. However, the lack of differences between groups suggests this was not a confounding factor. It was not possible to consistently measure the motion of C7 and T1 due to radiographic imposition. Disc degeneration was detected at C6/7 and C7/T1 and it is possible that differences in motion sharing inequality or variability (MSI or MSV) existed at these levels but remain undetected, albeit prevalence of degeneration at these levels were not different between groups. It is possible that the pattern of disc degeneration is discriminating in neck pain as is suggested in low back pain [36,37] however this study was not adequately powered to explore that.

## 5. Conclusions

This study found a moderate relationship between age and cervical spine motion sharing variability (MSV) in patients with neck pain but not in healthy controls during standardized, active weightbearing motion. Differences were not detected, however, in the prevalence or degree of cervical spine intervertebral disc degeneration or intervertebral motion sharing (MSI or MSV) between patients with mild-moderate neck pain and healthy controls.

## Figures and Tables

**Figure 1 jfmk-09-00055-f001:**
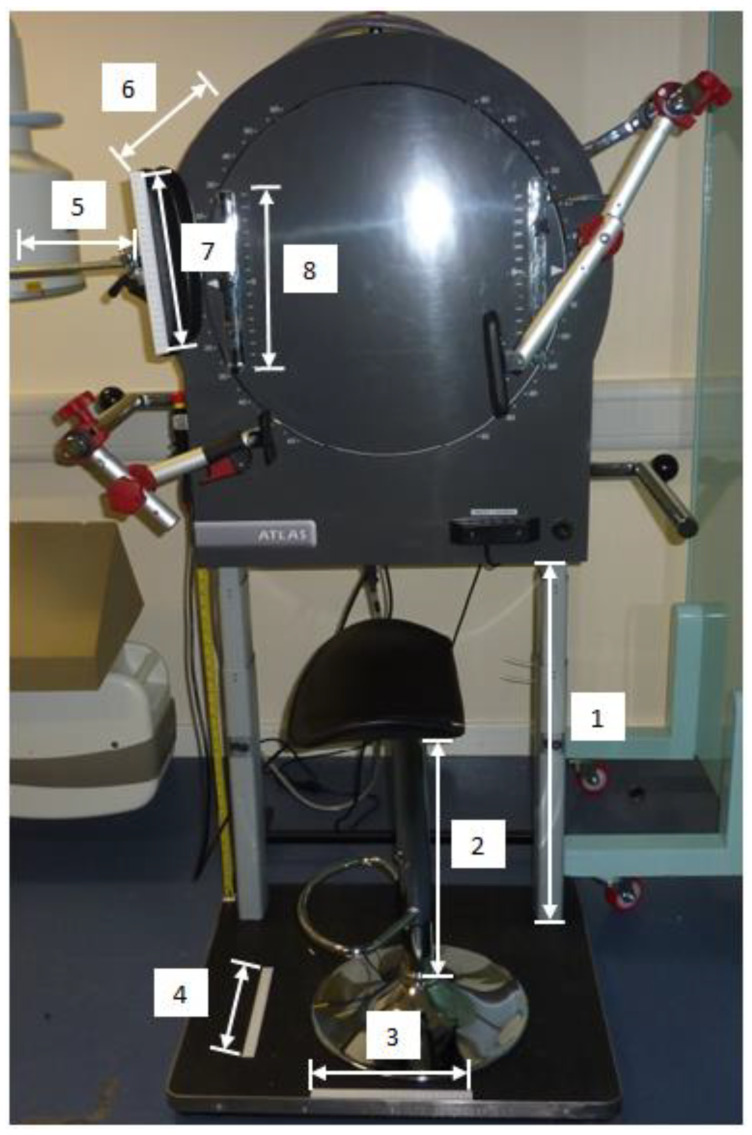
Stabilisation and motion-frame with aspects that were measured. 1—height of motion frame, 2—height of stool, 3—position of stool base, 4—position of stool base, 5—horizontal distance of face-rest, 6—distance from motion-frame to face-rest, 7—position of participant’s face on face-rest, 8—height of face-rest.

**Figure 2 jfmk-09-00055-f002:**
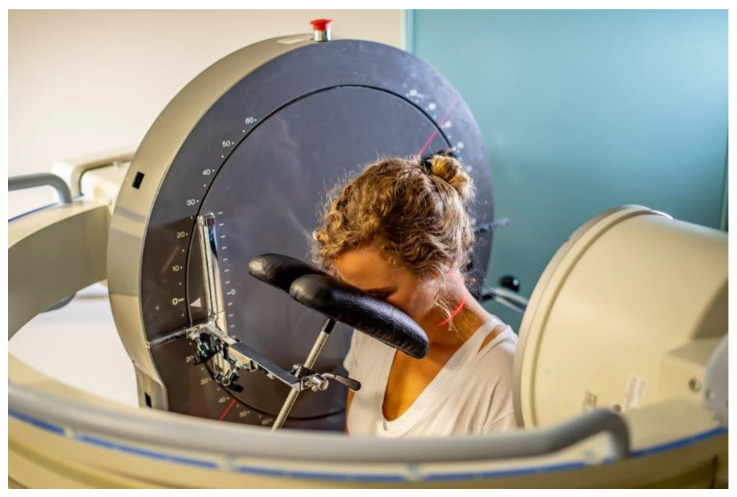
Example of following the face rest during flexion motion acquisition.

**Figure 3 jfmk-09-00055-f003:**
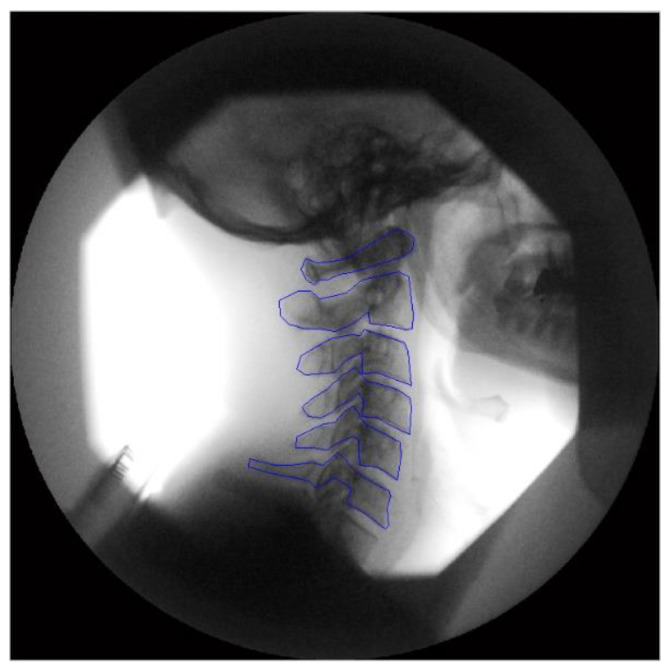
Tracking templates positioned on the first image from a fluoroscopic sequence.

**Figure 4 jfmk-09-00055-f004:**
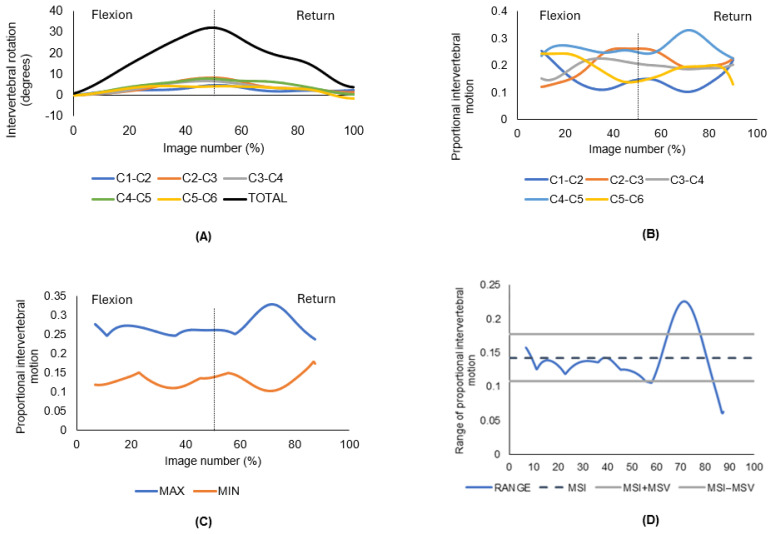
Derivation of motion-sharing inequality (MSI) and motion-sharing variability (MSV) from one representative participant’s QF image sequence obtained during cervical flexion and return. (**A**) Absolute intervertebral rotations, where flexion is considered an increase in intervertebral angle. The dotted line indicates maximum flexion. (**B**) The maximum and minimum ranges of proportional intervertebral motion. The dotted line indicates maximum flexion. (**C**) Proportional intervertebral rotations. The dotted line indicates maximum flexion. (**D**) MSI is the mean of the range of proportional intervertebral motion, MSV is the standard deviation of the range.

**Table 1 jfmk-09-00055-t001:** Characteristics and CDD in patients with neck pain and healthy controls.

	Patients (*n* = 29)	Healthy Controls (*n* = 30)	*p*
Age, mean (SD)	40 years (13.1)	41 years (13.1)	0.72 ^†^
Female, *n*	21	21	>0.99 ^‡^
Mean (SD) regional cervical spine ROM (°)	Flexion 49 (6.7)	53 (7.2)	0.04 ^†^
	Extension 51 (7.2)	56 (6.6)	0.03 ^†^
CDD/16, median (25, 75)	2 (1, 4)	2 (1, 3)	0.94 ^††^
Mean (SD) NRS score	5 (1.5)	-	-
Mean (SD) NDI score	13 (6.7)	-	-
Mean (SD) EQ-5D 5L VAS	75 (15.5)	-	-
Mean (SD) EQ-5D 5L Index	0.74 (0.09)	-	-

NRS, Numerical Rating Scale (0–10) for pain; NDI, Neck Disability Index (0–50); EQ-5D 5L, Euroquol; VAS, Visual Analogue Scale (0–100); ^†^ (unpaired) *t*-test; ^‡^ Fisher’s exact test; ^††^ Mann–Whitney U test.

**Table 2 jfmk-09-00055-t002:** Prevalence of disc degeneration by level in patients with neck pain and healthy controls.

	Intervertebral Ranges of Motion				Number of Degenerated Intervertebral Discs (K-L 1 or Above)		Number of Degenerated Intervertebral Discs (K-L ≥ 2)	
	Flexion, Degrees Mean (SD)		Extension, Degrees Mean (SD)					
Level	Patients	Healthy Controls	Patients	Healthy Controls	Patients	Healthy Controls	Patients	Healthy Controls
C2/3	6 (3.1)	6 (3.5)	4 (3.2)	5 (3.7)	16	13	2	0
C3/4	7 (3.8)	7 (2.8)	7 (3.7)	8 (5.5)	5	6	1	1
C4/5	6 (2.8)	6 (3.4)	8 (4.7)	11 (5.8)	7	5	1	3
C5/6	5 (2.9)	6 (3.9)	9 (4.9)	8 (4.9)	14	11	5	4
C6/7	-	-	-	-	11	12	4	7
C7/T1	-	-	-	-	9	8	0	0
Total	24 (8.9)	25 (11.3)	28 (12.5)	32 (16.0)	62	55	13	15

K-L, Kellgren–Lawrence Scale.

**Table 3 jfmk-09-00055-t003:** MSI and MSV in patients with neck pain and healthy controls.

		Patients (*n* = 29)	Healthy Controls (*n* = 30)	*p* ^††^
Flexion, median (25, 75)	MSI	0.34 (0.28, 0.41)	0.32 (0.24, 0.41)	0.38
	MSV	0.14 (0.09, 0.19)	0.13 (0.08, 0.21)	0.75
Extension, median (25, 75)	MSI	0.41 (0.32, 0.52)	0.36 (0.29, 0.45)	0.28
	MSV	0.10 (0.06, 0.21)	0.09 (0.06, 0.16)	0.34

The 25th and 75th percentiles (25, 75); MSI, motion-sharing inequality—is the mean of the range of proportional intervertebral motion and represents the unevenness of restraint between segments; MSV, motion-sharing variance—is the variance of the range of proportional intervertebral motion and represents the unevenness of control; ^††^ Mann–Whitney U test.

**Table 4 jfmk-09-00055-t004:** Correlations between age, CDD, MSI, and MSV in patients and healthy controls.

		Patients						Healthy Controls					
		CDD		MSI		MSV		CDD		MSI		MSV	
		r_s_	*p*	r_s_	*p*	r_s_	*p*	r_s_	*p*	r_s_	*p*	r_s_	*p*
Flex	Age	0.29	0.13	−0.05	0.79	−0.16	0.40	**0.52**	**<0.001**	−0.10	0.62	−0.21	0.27
	CDD			0.23	0.23	−0.06	0.77			−0.25	0.18	−0.16	0.41
	MSI					**0.53**	**<0.001**					**0.44**	**0.02**
Ext	Age			0.24	0.21	**0.63**	**<0.001**			0.28	0.14	0.24	0.20
	CDD			−0.08	0.70	0.09	0.63			−0.19	0.32	0.10	0.59
	MSI					**0.39**	**0.04**					0.29	0.12

CDD, Composite Disc Degeneration Score; MSI, motion-sharing inequality—the mean of the range of proportional intervertebral motion and represents the unevenness of restraint between segments; MSV, motion-sharing variance—the variance of the range of proportional intervertebral motion and represents the unevenness of control. Significant correlations (*p* < 0.05) shown in bold; r_s_, Spearman’s correlation coefficient.

## Data Availability

All data analysed in this study are presented in the manuscript. Underlying data are available from the corresponding author on reasonable request.

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
