# Peer review of "Disc Degeneration and Cervical Spine Intervertebral Motion: A Cross-Sectional Study in Patients with Neck Pain and Matched Healthy Controls"

_jfmk, 2024, doi:10.3390/jfmk9010055_

Round 1

Reviewer 1 Report

Comments and Suggestions for Authors

Dear Authors, the manuscript is original and relevant, because comented about the motion of the cervical in patient with and without neck pain, I believe that the manuscript can be considered for publication after major review.

Author Response

Thank you Reviewer 1 for your insightful and helpful feedback which have definitely improved the quality of our paper. We hope you agree. Please see our response below and please see attached revised paper if you wish to check on any of the changes.

Author Comments 

Response 

Change, page number 

Reviewer # 1 

Title: The word Sharing: does it mean balance between the segments of the cervical region? If so, I suggest rewriting the title to make it easier for the reader to understand. 

It refers to the proportion of motion that each segment contributes to the overall motion of the cervical spine. Since it might be ambiguous we have removed the word ‘sharing.’  

Title: Disc degeneration and cervical spine intervertebral motion sharing: A cross-sectional study in patients with neck pain and matched healthy controls, page 1. 

Abstract: Improve the description of your results, add data. 

We have added data as shown in the changes. [The MSI and MSV data have not been included as they are less familiar or intuitive and we wish to adhere to JFMK’s guidance on abstract length].  

Age was moderately correlated with MSV in cervical spine extension in patients only (r = 0.63, p<0.001). There were no significant differences in the prevalence of disc degeneration (CDD) between patients, who had on average mild pain and related disability, and healthy controls (median CDD 2 both groups, p =0.94). 

Introduction: Improve the introduction to better support the study. Add studies that evaluate ROM and functionality of the cervical region with the instruments that were used in your study 

To our knowledge we have included all studies that have evaluated the cervical region using the instruments used in this study. We would be humbled if you could let us know if we missed any and we would of course make any necessary changes.  

- 

Methods: Add figures of the tasks that were completed. 

Figures have been added to show the configuration of the stabilization and motion frame, and tracking templates on first fluoroscopic image.  

In brief, participants were seated with bracing rods comfortably positioned against the mid-thoracic spine and the sternum to promote a fixed trunk position (Figure 1), page 4-5. 

Participants performed flexion as the face rest rotated downwards, and extension as it moved upwards, keeping their faces’ gently in touch with the face rest throughout the motion (Figure 2), page 5.  

Manual registration was used on the first image to identify the individual vertebral bodies (Figure 3) then in all subsequent frames of the motion sequence these vertebrae were tracked with bespoke tracking codes written in Matlab (V2013 – The Mathworks Inc.), Page 6. 

What is the degree of degeneration of the volunteers in the neck pain group? How were you evaluated, did you use radiographic images? 

The degree of degeneration in the healthy volunteer group is provided in table 1 (composite disc degeneration score) and in table 2 (number of degenerated discs). Degeneration was identified from the initial image of each flexion imaging sequence as described on page 8. 

- 

In relation to the healthy group, what is the CDD score in table 1. Does this mean that the healthy group also had degeneration? 

Yes, the healthy group also had disc degeneration as was expected.  

- 

Wouldn't it be better for your analyzes to subgroup and take into account the degree of degeneration? Or subgroup by age? 

The sample size was not large enough to perform subgroup analysis otherwise we agree this would have been of value.  

- 

Results: I suggest redoing and changing the graphics designs 

There are no graphics in the results section. Perhaps you were referring to Figure 1 (now Figure 4) which we agree could have been clearer and has been re-done.  

Figure 4 refreshed, page 8. 

Discussion: Some paragraphs could be rewritten. Very long paragraphs that are difficult to understand 

We have reconfigured into shorter paragraphs to improve readability.  

Shorter paragraphs, pages 11-13. 

I suggest inserting new citations to improve the discussion with your data. 

The citations included in the discussion are contemporaneous. However, we will of course consider any suggested additions. 

- 

Conclusion: It could be more direct and just answer your objectives 

We have reduced the conclusion as advised.  

This study found a moderate relationship between age and cervical spine motion sharing variability (MSV) in patients with neck pain but not in healthy controls during standardized, active weightbearing motion. Differences were not detected, however, in the prevalence or degree of cervical spine intervertebral disc degeneration or intervertebral motion sharing (MSI or MSV) between patients with mild-moderate neck pain and healthy controls. In future studies QF could be combined with simultaneous EMG recording to elucidate the relationship between neck muscle activity and cervical intervertebral motion. Researchers should also aim to measure intervertebral motion passively to assess the role of the passive restraining elements in neck pain in the absence of muscle stabilisation., page 13.  

Reviewer 2 Report

Comments and Suggestions for Authors

Review jfmk-2866032-peer-review-v1

The paper Disc Degeneration and Cervical Spine Intervertebral Motion Sharing: A Cross-Sectional Study in Patients with Neck Pain 3 and Matched Healthy Controls is interesting, but the way of presentation requires improvement. It is necessary to clarify and refine some elements. Below, you will find these comments listed not in order of importance but in the order of occurrence. Kindly consider these remarks to enhance your paper accordingly.

Abstract. The information in lines 13-19 should be abbreviated because they are similar. The abstract lacks a written objective and there is no conclusion. Please improve this part.

The introduction is well-written. In my opinion, it is recommended to omit the last sentence containing the hypothesis, as subsequent authors do not refer to it.

Material and Methods. I suggest organizing chapters 3, 4, and 5 as subchapters.

In Chapter 3, there should be information regarding pain medications. Was this information included in the exclusion criteria? Furthermore, on lines 93-94, it would be beneficial to include links (citation) to the Numerical Rating Scale, Neck Disability Index, and EuroQuol-5D-5L, along with brief descriptions of each.

Chapter 4 should be divided into two sections: the first focusing on Image Acquisition, and the second on Image Analysis.

Line 101 – The acronym QF should be explained again.

After lines 101-116, it is recommended to include some pictures of the equipment and patients using the equipment.

Line 117 – Please use '*.dicom file' instead of 'DICOM file’.

Line 122 – based on which “The accuracy of this method has been determined as 0.21° in flexion and 0.34° in extension”? Please include an explanation in the text.

Line 124 – ICC and SEM abbreviations should be explained.

Lines 129 – 130 – I don't understand the concepts “motion sharing inequality (MSI) and motion sharing variability (MSV)”, please clarify in the text.

The quality of Figure 1 is poor; the font is too small, making it difficult to read. It's unclear whether these are means or examples and whether the data pertains to healthy individuals or patients. Additionally, the lack of units on the vertical axes is a concern. The figure needs clarification on where flexion and return points are located. It's essential to compare data between healthy individuals and patients. This element must be revised for better clarity and accuracy.

Linie 142 – I do not understand what is proportional intervertebral motion across the N images of the motion sequence. Please also explain how much N was.

Line 151 - the Kellgren and Lawrence rating scale – please give a brief description.

Line 154 – What do the points 0 - 4 mean?

Lines 160 – 167 – Please provide a detailed description of the variables considered, specify the comparisons made, and outline the tests conducted.

Line 162 – How do I understand the distributions of continuous data?

Results. Please write the p-value uniformly throughout the paper, namely 4 decimal places.

Table 1 - description of the table Characteristics ... of what? I do not understand why there is a notation for women 21 and in parentheses (70) - What is it? As a percentage, I do not understand what and what this information is supposed to add. Why there are no years next to the age? I do not understand the notation for Mean (SD) pain score/10, Mean (SD) NDI score/50, Mean (SD) EQ-5D 5L VAS/100. What does it mean /10, /50 or VAS/100?

Table 2 – In the table, what is in the columns? Patients, their number? Or the values of the angles? Why are there no units? Why is there no SD for the last row?

Table 3 – In the table, what is (25, 75)? Why are there no units? P - value is the result of what test?

Table 4 – why is there no explanation of rs parameters anywhere? Please also, in the material and methods section, specify ranges for correlations, defining when they are weak, moderate, or strong.

Please add the limitations of the study.

Author Response

Dear Reviewer 2, thank you very much for your insightful and helpful comments which have hugely improved our paper. We hope you agree. Please see below for our response to your comments and please find attached the revised paper should you wish to check any of the changes.

Reviewer # 2 

 Response

 Change, page number

Abstract. The information in lines 13-19 should be abbreviated because they are similar. The abstract lacks a written objective and there is no conclusion. Please improve this part. 

We have made changes following your advice to enhance the abstract. Further text has not been added to respect the journal author guideline on the length of the abstract.   

The objective was to determine if average age, CDD, MSI, and MSV values were correlated and if there were differences in these variables between the neck pain group and the healthy control group, page 2. 

The introduction is well-written. In my opinion, it is recommended to omit the last sentence containing the hypothesis, as subsequent authors do not refer to it. 

The sentence relating to hypothesis has been removed.  

Final sentence of introduction removed, page 3.  

Material and Methods. I suggest organizing chapters 3, 4, and 5 as subchapters. 

We are not clear exactly what is being suggested here. We hope that the changes that have been made conform to your suggestion.  

- 

In Chapter 3, there should be information regarding pain medications. Was this information included in the exclusion criteria? 

Patients with at least two weeks mechanical neck pain rated 3 or more on the 11-point NRS were included. They were permitted to continue with any pain medication they were taking and were not excluded on this basis.  

“Patients were not excluded on the basis of any pain medication they were taking” has been added to the Participants section, page 4. 

Furthermore, on lines 93-94, it would be beneficial to include links (citation) to the Numerical Rating Scale, Neck Disability Index, and EuroQuol-5D-5L, along with brief descriptions of each. 

Citations and short descriptions have been added for each instrument.  

Patients needed to have at least two weeks’ mechanical neck pain rated 3 or more on the 11-point Numerical Rating Scale15 as a measure of pain severity, to be included. Patients were not excluded on the basis of any pain medication they were taking.  

Patients also completed the Neck Disability Index16 as a measure of pain-related disability and EuroQuol-5D-5L17 a quality of life measure. 

Chapter 4 should be divided into two sections: the first focusing on Image Acquisition, and the second on Image Analysis. 

This part has been divided into two sections as advised.  

Image acquisition section is pages 4-6, image analysis section is pages 6 – 9.  

Line 101 – The acronym QF should be explained again. 

This has been added.  

The quantitative fluoroscopy (QF) acquisition procedure, page 4. 

After lines 101-116, it is recommended to include some pictures of the equipment and patients using the equipment. 

Please see addition of 3 figures.  

Figure 1, page 5; Figure 2, page 6, Figure 3, page 7. 

Line 117 – Please use '*.dicom file' instead of 'DICOM file’. 

Respectfully it would appear the form ‘DICOM’ is standard in the published literature, for example, Larobina (2023). Please kindly advise if we have misinterpreted the request.  

Larobina, M., 2023. Thiry Years of the DICOM Standard. Tomography 9(5), 1829-1838.  

- 

Line 122 – based on which “The accuracy of this method has been determined as 0.21° in flexion and 0.34° in extension”? Please include an explanation in the text. 

Explanatory text has been added.  

The accuracy and repeatability of this method was determined in an earlier calibration study using a model consisting of a moveable human cervical segment (C4-5) with a digital inclinometer attached. Motion measured by the digital inclinometer was compared to that measured by QF. Accuracy was reported, page 7.  

Line 124 – ICC and SEM abbreviations should be explained. 

New sentences have been added to explain the abbreviations.  

Accuracy was reported as 0.21° in flexion and 0.34° in extension9. For repeatability, intraclass correlation coefficients (ICC) and the standard error of measurement (SEM) were calculated19, page 7. 

Lines 129 – 130 – I don't understand the concepts “motion sharing inequality (MSI) and motion sharing variability (MSV)”, please clarify in the text. 

The purpose of figure 4 (figure 1 in the original submission) and accompanying text is to explain the concepts of MSI and MSV. We hoped the improved figure and text helps to clarify these.  

Please see change below.  

The quality of Figure 1 is poor; the font is too small, making it difficult to read. It's unclear whether these are means or examples and whether the data pertains to healthy individuals or patients. Additionally, the lack of units on the vertical axes is a concern. The figure needs clarification on where flexion and return points are located. It's essential to compare data between healthy individuals and patients. This element must be revised for better clarity and accuracy. 

This figure (now figure 4) has been recreated to make it easier to read. Hopefully it is much clearer now that this is the example from one participant as a visualisation of how these parameters were derived. A line has been included to make the maximum more obvious to show flexion and return.  

New text: . Figure 4 provides an illustration of how MSI and MSV were calculated, for all participants, with the example of one representative QF image sequence obtained from one participant during cervical flexion and return (Figure 4A-D), page 7. 

Figure 4 has been recreated with much greater clarity of colour and text, page 8.  

New text: 

  In Figure 4A are the absolute intervertebral rotations in degrees (y-axis) shown against the corresponding image from the sequence as a percentage (x-axis) to best indicate flexion and return. The black line represents the total range contributed to by all segments. Figure 4B shows the proportion contributed to the total range by each segment. The maximum and minimum proportional intervertebral motion at each image is then shown in Figure 4C.  The range between each maximum and maximum is calculated and from that, the mean of the range of proportional intervertebral motion as shown in Figure 4D. The mean of that range is MSI (motion sharing inequality), the standard deviation of that range is MSV (motion sharing variability), page 9.  

Linie 142 – I do not understand what is proportional intervertebral motion across the N images of the motion sequence. Please also explain how much N was. 

We hope this is clearer from the improved figure and text above. Some text also added to explain how much N typically was.  

where fRCi is the average of the range of proportional intervertebral motion across the (N) images of the motion sequence (Figure 4D). The mean (standard deviation) number of images per sequence (flexion and extension) across all participants was 290 (38) images.   

Line 151 - the Kellgren and Lawrence rating scale – please give a brief description. 

A brief description has been provided.  

This scale is a common method of classifying the severity of osteoarthritis from imaging providing a score from 0 (no radiological findings of osteoarthritis) through to 4 (large osteophytes, marked joint space narrowing, severe sclerosis and definite deformity of bone contour). 

Line 154 – What do the points 0 - 4 mean? 

Please see response above.  

- 

Lines 160 – 167 – Please provide a detailed description of the variables considered, specify the comparisons made, and outline the tests conducted. 

A more detailed description is warranted as you indicate. To make it easier for the reader, rather than add this here we have improved the legends provided for each data table showing which statistical test was used for which variable. We hope this adequately addresses the request. 

Legend for Table 3 now includes: ††, Mann-Whitney U test 

Legend for Table 3 now includes: rs, Spearman’s correlation coefficient 

Line 162 – How do I understand the distributions of continuous data? 

In the results section normally distributed data is reported as mean (standard deviation), while non-normal data is reported as median (25th percentile, 75th percentile). 

- 

Results. Please write the p-value uniformly throughout the paper, namely 4 decimal places. 

Thank you for point this out - we were incorrect to use three decimal places in cases where we provided p=0.000. Rather than use 4 decimals places, we respectfully wish to use the statistical reporting guideline used by both the New England Journal of Medicine and the Journal of the American Medical Association (Aguinis et al 2021). 

Using this approach, p-values larger than p=0.01 are reported to two decimal places, those between 0.01 and 0.001 reported to three decimal places, and those smaller than 0.001 reported as p<0.001.  

Aguinis, H., Vassar M., and Wayant, C., 2021. On reporting and interpreting statistical significance and p values in medical research. BMJ Evidence Based Medicine 26(2), 39-42.  

P values have been corrected throughout to reflect recommended statistical reporting.  

Table 1 - description of the table Characteristics ... of what? I do not understand why there is a notation for women 21 and in parentheses (70) - What is it? As a percentage, I do not understand what and what this information is supposed to add. Why there are no years next to the age? I do not understand the notation for Mean (SD) pain score/10, Mean (SD) NDI score/50, Mean (SD) EQ-5D 5L VAS/100. What does it mean /10, /50 or VAS/100? 

The percentage is supposed to indicate the proportion of the cohort who was female. We agree this is perhaps not helpful as percentages should only strictly be used for numbers larger than 100.  

The notion is not as clear as we intended. The /10, /50 represents “out of” for example, pain score/10 means pain score “out of” 10. These “out of” scores have been added to the table legend instead.  

Percentages have been removed, years added for age, and confusing notation removed, from table 1.  

Table 2 – In the table, what is in the columns? Patients, their number? Or the values of the angles? Why are there no units? Why is there no SD for the last row? 

We neglected to include the units where needed, and the SDs were missed from the last row in error. Thank you for spotting this.  

Units and missing SDs added to Table 2.  

Table 3 – In the table, what is (25, 75)? Why are there no units? P - value is the result of what test? 

(25, 75) refers to 25th and 75th percentiles showing the range around the median.  

We have included this information in the table 3 legend. 

Table 4 – why is there no explanation of rs parameters anywhere? Please also, in the material and methods section, specify ranges for correlations, defining when they are weak, moderate, or strong 

Apologies for this oversight, now corrected.  

We have included rs information in the table 4 legend. Ranges added to materials and methods section:  

Correlations were made with the Spearman’s correlation coefficient and interpreted as follows: 0.00 – 0.10, negligible; 0.10 – 0.39, weak; 0.40 – 0.69, moderate; 0.70 – 0.89, strong, 0.90 – 1.00, very strong. 

Reviewer 3 Report

Comments and Suggestions for Authors

Dear authors,

The manuscript is interesting but I suggest some improvements before it can be published.

Title: It is not an attractive title, try to improve it.

Abstract:

Line 13 to 19: this introduction of the topic is too long, try to reduce it.

Line 20-21: Indicate the socio-demographic variables, the number of women and men as well as the age values in brackets.

Line 26 You should indicate in more detail the statistical analysis used.

Keywords: Indicate the maximum number of Keywords allowed by the journal.

Introduction: Line 54. The reference to the statement ending this sentence is missing.

Line 75-78: separate this from the previous paragraph, the objective and hypothesis from the rest.

Material and methods.

Line 86: Please indicate the socio-demographic variables in brackets and their standard deviation.)

Line 88: Please indicate in a separate paragraph the inclusion and exclusion criteria. Please indicate if these criteria are the same as in other studies with the corresponding citation.

Line 99- Indicate if the study was registered on a public platform and the registration number.

Line 111- Indicate if the mean instruments are validated and their validation reference or if they have been used in another study.

Line 160-166. Explain in more detail the statistical analysis performed, why you used the kappa and not the standard error of measurement.

You should indicate the number of measurements recorded for each variable.

Could you add a table with the results of the reliability analysis performed?

Discussion.

This section is too long. Shorten it

The paragraphs are too few and it joins ideas that should be in different paragraphs. Please separate it.

Line 285. Add a section on the limitations of the study and then add a paragraph on the future lines of research.

Conclusions.

Line 293-add this to the discussion.

Author Response

Dear Reviewer 3, thank you very much for your insightful and helpful comments which have definitely helped to strengthen our paper. We hope you agree. Please see our responses to your comments below, and please find attached the revised paper should you wish to check any of the changes. 

Reviewer # 3 

Title: It is not an attractive title, try to improve it. 

We have removed the word ‘sharing’ as that seemed to cause some confusion.  

Title: Disc degeneration and cervical spine intervertebral motion sharing: A cross-sectional study in patients with neck pain and matched healthy controls, page 1. 

Abstract: 

Line 13 to 19: this introduction of the topic is too long, try to reduce it. 

We have removed some words to reduce topic introduction length.  

The second sentence of the abstract has been deleted, page 1.  

Line 20-21: Indicate the socio-demographic variables, the number of women and men as well as the age values in brackets. 

We respectfully have been unable to do this due to the word count limit on the abstract as stipulated by JFMK. However, this information is provided in the paper.    

- 

Line 26 You should indicate in more detail the statistical analysis used. 

We have added some data to the abstract. More words are difficult du to respecting the word count limit.  

Age was moderately correlated with MSV in cervical spine extension in patients only (r = 0.63, p<0.001). There were no significant differences in the prevalence of disc degeneration (CDD) between patients, who had on average mild pain and related disability, and healthy controls (median CDD 2 both groups, p =0.94). 

Keywords: Indicate the maximum number of Keywords allowed by the journal. 

Thank you for pointing out this omission.  

Keywords: disc degeneration, neck pain, fluoroscopy – added under abstract, page 2 

Introduction: Line 54. The reference to the statement ending this sentence is missing. 

We respectfully have not been able to identify the sentence in the introduction that has a reference missing. Our line numbering is perhaps different. We have tried hard to find this.  

- 

Line 75-78: separate this from the previous paragraph, the objective and hypothesis from the rest. 

We have separated as advised.  

End of introduction is now a separate paragraph, page 4.  

Material and methods. 

Line 86: Please indicate the socio-demographic variables in brackets and their standard deviation.) 

Line 88: Please indicate in a separate paragraph the inclusion and exclusion criteria. Please indicate if these criteria are the same as in other studies with the corresponding citation. 

There were no socio-demographic variables collected beyond sex and age.  

The inclusion criteria are the same as in the study with the corresponding citation. 

Inclusion and exclusion criteria have been presented in a separate paragraph, page 4. 

Line 99- Indicate if the study was registered on a public platform and the registration number. 

The study was not registered on a public platform.  

- 

Line 111- Indicate if the mean instruments are validated and their validation reference or if they have been used in another study. 

Citations have been added for the data collection instruments which have all been validated.  

Patients needed to have at least two weeks’ mechanical neck pain rated 3 or more on the 11-point Numerical Rating Scale15 as a measure of pain severity, to be included. Patients were not excluded on the basis of any pain medication they were taking.  

Patients also completed the Neck Disability Index16 as a measure of pain-related disability and EuroQuol-5D-5L17 a quality of life measure. 

Line 160-166. Explain in more detail the statistical analysis performed, why you used the kappa and not the standard error of measurement. You should indicate the number of measurements recorded for each variable. 

Kappa was used where the data was ordinal (as in the 0-4 Kellgren and Lawrence Scale). SEM would normally be used in context of continuous data hence why it was not used.  

- 

Could you add a table with the results of the reliability analysis performed? 

This is a good suggestion and we have discussed this. We feel this risks the paper being too long and we do not want to detract from the main thrust of the paper. The reliability analysis was performed only to improve confidence of the results from the main study.   

- 

Discussion. 

This section is too long. Shorten it 

The discussion section has been better organized.  

- 

The paragraphs are too few and it joins ideas that should be in different paragraphs. Please separate it. 

The discussion section has split into more paragraphs. Thank you, it helps readability.  

- 

Line 285. Add a section on the limitations of the study and then add a paragraph on the future lines of research. 

This is the typical approach, of course, but we found that including future lines of research in connection with the related limitation(s) of the study worked quite well.  

- 

Conclusions. 

Line 293-add this to the discussion. 

We have done this, thank you. 

This study found a moderate relationship between age and cervical spine motion sharing variability (MSV) in patients with neck pain but not in healthy controls during standardized, active weightbearing motion. Differences were not detected, however, in the prevalence or degree of cervical spine intervertebral disc degeneration or intervertebral motion sharing (MSI or MSV) between patients with mild-moderate neck pain and healthy controls. In future studies QF could be combined with simultaneous EMG recording to elucidate the relationship between neck muscle activity and cervical intervertebral motion. Researchers should also aim to measure intervertebral motion passively to assess the role of the passive restraining elements in neck pain in the absence of muscle stabilisation., page 13.  

Round 2

Reviewer 2 Report

Comments and Suggestions for Authors

The authors have satisfactorily improved the paper. I have no further comments.